# Tree Species Classification of Drone Hyperspectral and RGB Imagery with Deep Learning Convolutional Neural Networks

**Somayeh Nezami [1], Ehsan Khoramshahi [1,2,*] , Olli Nevalainen [3] , Ilkka Pölönen [4] and Eija Honkavaara [1]**

[1] Department of Remote Sensing and Photogrammetry of the Finnish Geospatial Research Institute FGI, Geodeetinrinne 2, FI-02430 Masala, Finland; snezami89@gmail.com (S.N.); Eija.Honkavaara@nls.fi (E.H.)
[2] Department of Computer Science, University of Helsinki, FI-00560 Helsinki, Finland
[3] Finnish Meteorological Institute, Climate System Research, 00560 Helsinki, Finland; olli.nevalainen@fmi.fi
[4] Faculty of Information Technology, University of Jyväskylä, Mattilanniemi 2, 40014 Jyväskylä, Finland; ilkka.polonen@jyu.fi
* Correspondence: ehsan.khoramshahi@nls.fi; Tel.: +358404444135

**Abstract:** Interest in drone solutions in forestry applications is growing. Using drones, datasets can be captured flexibly and at high spatial and temporal resolutions when needed. In forestry applications, fundamental tasks include the detection of individual trees, tree species classification, biomass estimation, etc. Deep neural networks (DNN) have shown superior results when comparing with conventional machine learning methods such as multi-layer perceptron (MLP) in cases of huge input data. The objective of this research is to investigate 3D convolutional neural networks (3D-CNN) to classify three major tree species in a boreal forest: pine, spruce, and birch. The proposed 3D-CNN models were employed to classify tree species in a test site in Finland. The classifiers were trained with a dataset of 3039 manually labelled trees. Then the accuracies were assessed by employing independent datasets of 803 records. To find the most efficient set of feature combination, we compare the performances of 3D-CNN models trained with hyperspectral (HS) channels, Red-Green-Blue (RGB) channels, and canopy height model (CHM), separately and combined. It is demonstrated that the proposed 3D-CNN model with RGB and HS layers produces the highest classification accuracy. The producer accuracy of the best 3D-CNN classifier on the test dataset were 99.6%, 94.8%, and 97.4% for pines, spruces, and birches, respectively. The best 3D-CNN classifier produced ~5% better classification accuracy than the MLP with all layers. Our results suggest that the proposed method provides excellent classification results with acceptable performance metrics for HS datasets. Our results show that pine class was detectable in most layers. Spruce was most detectable in RGB data, while birch was most detectable in the HS layers. Furthermore, the RGB datasets provide acceptable results for many low-accuracy applications.

**Keywords:** deep learning; drone imagery; hyperspectral image classification; tree species classification; 3D convolutional neural networks

## 1. Introduction

Automating tree species classification has been widely considered as an important task in forest science, since automatic detection of tree species significantly decreases the requirement for manual work in forest inventory [1]. Among the modern techniques that are employed for tree species detection, airborne remote sensing technologies have recently been employed at an unprecedented rate. This technology has provided the capability to quickly measure and classify a vast number of trees.

Unmanned aerial vehicles (UAVs) have become a central part of airborne remote sensing. Tree species classification by UAV remote sensing initially involves the process of selecting appropriate sensors that provides relevant features of trees. Any set of observable features of a tree, such as overall shape or color intensity, leaf properties, spectral response of a tree to electromagnetic spectrum, or geometric aspects of tree tops, such as tree height model from a dense point cloud, are valuable for classifying species of individual trees. The subsequent step is to make observations of these features, and finally, machine learning classifiers are used to investigate the ability of the features in separating different tree species.

Investigating tree crowns with different data sources such as color data (RGB), hyperspectral (HS) data, or point clouds is of high interest to the remote sensing and forest research communities. Nowadays, RGB imaging is the most basic and affordable data source for many remote sensing solutions. We may safely label it as the most accessible UAV data layer. On the other hand, HS data was long considered a relatively expensive data. Applications of HS imaging (his) in classification tasks have been successfully demonstrated in many research works, e.g., [2–6]. Hyperspectral images (HSI) have been gradually transformed into a more accessible layer of data. As an expected result, it has been overly employed in recent forest applications, e.g., [7–10]. Employing HSI in a tree species classification requires radiometric processing of HSIs, which is followed by feature extraction and selection, and classification aspects. Radiometric processing refers to the process of tuning HSIs through an optimization process that ensures the radiometric consistency of the data. This process improves the repeatability of the classification in different illumination conditions and study areas [11]. The output of radiometric processing is a reflectance mosaic that has been produced from HSIs in different capturing conditions. Feature selection in this context refers to the problem of selecting the most relevant information from a reflectance mosaic for the classification problem. This step could be an automatic or a manual process [4]. In many tree classification problems, only geometric features have been employed in the classification step since spectral information was hard to acquire or expensive to include. However, HSIs contain both spectral and geometric information as a unified dataset. In this context, the geometric features of an object are combined with the spectral information to improve classification metrics.

Usually the feature selection is mixed up with choosing a suitable classifier. Classification aspects refer to the process of selecting an appropriate classifier, tuning, training, and validating it, and investigating its repeatability. It also includes the technicalities that are involved in the usage of the trained classifier. Many supervised and unsupervised classification methods are eligible for addressing an HSI classification problem. To name a few, we can mention logistic regression (LR), probabilistic graphical models (PGM), decision tree classifiers, support vector machines (SVM), nearest-neighbor classifiers (NN), clustering methods such as k-means, deep learning methods such as multi-layered perceptron (MLP), and convolutional neural networks (CNN), etc. [3,12]. The aforementioned classification techniques have specific behavior, benefits, and drawbacks that have been discussed by many researchers for tree species classification problems [2,4,7,9,10,13,14].

In recent HSI classification studies, 3D-CNN models have demonstrated superior performance over conventional methods such as SVM, KNN, or other deep-learning-based HSI classification methods such as deep belief network-logistic regression (DBN-LR), stacked auto encoder-logistic regression (SAE-LR), and 2D-CNN [4,10]. In case of big datasets with high spatial and spectral resolution, CNNs seem to be one of the best classifiers [5]. In particular, 3D-CNNs are of high interest because of their capabilities in addressing high-dimensional classification problems that other methods are less capable of handling.

The three main tree species in Finnish boreal forests are pine, spruce, and birch. Although the 3D shapes of coniferous pine and spruce crowns are usually clearly different (spruces being more conical [15]), their canopy-level spectra are quite similar [16]. Birch trees, however, often have more distinguishable canopy-level spectra, especially at near-infrared due to higher leaf-level reflectance at NIR and higher contribution of first-order scattering ([16,17]). Tree species classification by employing

HS data, lidar data, or a combination of these has been successfully demonstrated in many recent research works. For example, Korpela et al. [18] employed two airborne lidar sensors to classify major tree species in a forest area in Finland. They reported ~90% classification accuracy for Scot pine, Norway spruce, and birch by employing intensity variables. Dalponte et al. [19] employed and compared two HS sensors (HySpex-VNIR 1600 and HySpex-SWIR 320i) for tree species classification. They compared SVM and Random Forest (RF) to distinguish tree species among four classes: Norway spruce, Scots pine, scattered birch, and other broadleaves. They reported an overall classification accuracy of 90% by employing the first HS sensor (HySpex-VNIR 1600); however, they did not report any significant (>5%) difference between using SVM and RF. In this work, birch was reported as the least distinguishable class among all three species with a producer accuracy of 61.5%. Heinzel and Koch [20] employed full-waveform LIDAR for a six-class tree-species classification. Later, they investigated multiple data sources for tree species classification [21]. They employed features from multiple data sources including lidar, HS, and color infrared (CIR) to classify four tree species: pine, spruce, oak, and beech. They employed SVM as the high-dimensional classifier. Yao et al. [22] employed full waveform lidar data for tree species classification. They applied an unsupervised classification method to distinguish deciduous and coniferous trees. They reported 93%, and 95% classification accuracies for their proposed unsupervised and a supervised classification method respectively. Fassnacht et al. [23] reviewed some recent research on tree species classification. They highlighted the problem of locality of solutions and the essence for global methods that are able to address large geographical extents. Raczko and Zagajewski [13] compared three classifiers (MLP, SVM, and RF) for classifying four tree species by employing an airborne HS sensor. They reported 77% highest median overall classification for MLP. Yu et al. [1] employed multispectral airborne laser scanning (ALS) for individual tree detection and classification in a boreal forest area. They classified Scots pine, Norway spruce, and birch with an RF classifier. They reported 85.9% overall accuracy for combined point cloud and single channel intensity features. Franklin et al. [14] employed a rotating multispectral sensor on a UAV to classify four tree species in a northern hardwood forest. They reported an overall classification accuracy of 78% by employing RF classifier. Pölönen et al. [2] employed a 3D-CNN on RGB and HS data to classify three main tree species in a boreal forest in Finland. They demonstrated an overall accuracy of 96.2% on their validation dataset by employing their proposed 3D-CNN model. Ferreira et al. [24] employed WorldView-3 satellite multispectral sensor for tree species classification in a tropical forest area. They employed 16 bands from visible to near infrared and shortwave infrared. They classified eight tree species with SVM, and reported an average classification accuracy of between 60% and 96% for most tree species.

Thus far, most of the studies on tree species classification by HSIs have focused on feature-based machine learning techniques, such as RF and KNN [6,7]. There are also a few researchers who employed 3D-CNN in tree species classification by HSIs [2], or RGB images [8]. The objectives of this study were to develop and compare 3D-CNN models and features for classification of three major tree species in the boreal forest.

We developed 3D-CNN models using sample data from a test site in Finland using two types of feature and sensor combinations: a comprehensive multisensory system capturing HS and normal color (RGB) images, and photogrammetric 3D point clouds based on structure from motion, as well as a low-cost system based on RGB color features and 3D point clouds. We assessed the performance of the 3D-CNN classifier using an independent dataset and compared them to an MLP classifier. The study used partially the same dataset as used by Nevalainen et al. [7] and Tuominen et al. [25].

In general, we followed a similar approach as [2,4] in classification methodology. In feature selection, we completed studies such as [1,2,7]. The main contributions of this work are:

1. An efficient structure for a 3D-CNN network that is suitable for tree species classification is proposed and investigated. The proposed structure achieves a very high classification accuracy, while it is simpler than previously proposed structures [2];
2. An evaluation of the proposed model is performed by comparing it to an MLP classifier;

3. Different feature combinations originating from different potential sensors are compared to find the most relevant feature set.

This paper has the following structure. In Section 2 we describe the study site, the used dataset, and the methodology. In Section 3, we present the results, and the discussion and conclusions are given in Sections 4 and 5, respectively.

## 2. Materials and Methods

### 2.1. Test Site and Remote Sensing Datasets

The Vesijako research forest area in the municipality of Padasjoki in southern Finland (approximately 61°24′ N and 25°02′ E) was used as the study area. The area has been used as a research forest by the Natural Resources Institute of Finland. The same dataset has been used previously by Nevalainen et al. [7] in individual tree detection and tree species classification. Three sampling regions were selected in the area (Figure 1).

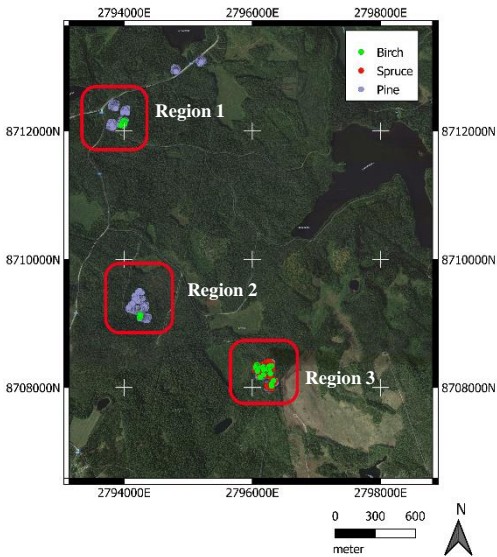

**Figure 1.** Location and distribution of all regions. Each class is individually colored.

A total of 11 test plots were captured in the Vesijako area using a small hexacopter UAV in eight separate flights between 25 and 26 June 2014. An HS camera based on a tunable Fabry-Pérot interferometer (FPI) [11,26–29] was used to capture the 2D frame-format HSIs. The image size was 1024 x 648 pixels, and the pixel size was 11 μm. The FPI camera has a focal length of 10.9 mm; the field of view (FOV) is ±18° in the flight direction, ±27° in the cross-flight direction, and ±31° at the format corner. Because of the sequential HS data capture mode of the individual bands (0.075 s between adjacent exposures, 1.8 s during the entire data cube with 24 exposures), each band of the data cube had a slightly different position and orientation, which had to be taken into account in the post-processing phase. In this study, a total of 33 bands were used with the full width of the half maximum (FWHM) of 11-31 nm (see Table 1). The UAV was also equipped with an ordinary RGB compact digital camera, the Samsung NX1000, in order to capture high spatial resolution photogrammetric imagery. The camera has a 23.5 x 15.7 mm CMOS sensor with 20.3 megapixels and a 16 mm lens.

**Table 1.** Spectral settings of the Fabry-Perot interferometer (FPI) camera at visible (VIS) and near-infrared (NIR) channels. L0: central wavelength; FWHM: full width at half maximum.

| |
| --- |
| **L0 (nm):** 507.60, 509.50, 514.50, 520.80, 529.00, 537.40, 545.80, 554.40, 562.70, 574.20, 583.60, 590.40, 598.80, 605.70, 617.50, 630.70, 644.20, 657.20, 670.10, 677.80, 691.10, 698.40, 705.30, 711.10, 717.90, 731.30, 738.50, 751.50, 763.70, 778.50, 794.00, 806.30, 819.70 |
| **FWHM (nm):** 11.2, 13.6, 19.4, 21.8, 22.6, 20.7, 22.0, 22.2, 22.1, 21.6, 18.0, 19.8, 22.7, 27.8, 29.3, 29.9, 26.9, 30.3, 28.5, 27.8, 30.7, 28.3, 25.4, 26.6, 27.5, 28.2, 27.4, 27.5, 30.5, 29.5, 25.9, 27.3, 29.9 |

The datasets were captured using a flying height of 83–94 m from the ground level. This resulted in an average ground sampling distance (GSD) of 8.6 cm for the FPI images and 2.3 cm for the RGB images on the ground level. The flight height was 62–73 m from the tree tops, thus, the average GSDs were 6.5 cm and 1.8 cm at tree tops for the FPI and RGB datasets, respectively. Imaging conditions were quite windless, but illumination varied a lot between cloudy and sunny in different flights.

Geometric processing included calculations of the interior orientation parameters of the sensors and the exterior orientation parameters of the images using the structure-from-motion technique, and measurement of the 3D object model. Agisoft PhotoScan Professional commercial software (AgiSoft LLC, St. Petersburg, Russia) was used for the major part of the processing. After calculating the image orientations, dense point clouds with a 5-cm point interval were generated using two-times down-sampled RGB images using the PhotoScan. From the dense point clouds, the canopy height models (CHM) were calculated by utilizing the digital terrain model (DTM) generated using the national airborne laser scanning (ALS) data [NLS]. The image orientations and point clouds were transformed to the ETRS-TM35FIN coordinate system using the GCPs in the area. The FGI's in-house C++ software was used for the band registration of the HSIs [11,26,28,29].

Radiometric processing was necessary to transform the HSIs captured under varying conditions to uniform reflectance mosaics. The processing included a laboratory-based calibration of the images [14] and elimination of non-uniformities caused by illumination differences in the individual images using the FGI's radiometric block adjustment software [11,26]. The radiometric model parameters were calculated separately for each band. The DNs were transformed to reflectance using the reflectance panels installed in the area. HS orthophoto mosaics were calculated with 10 cm GSD from the FPI images using the FGI's in-house mosaicking software radBA [11,26]. The RGB mosaics were calculated using the PhotoScan mosaicking module with a 5 cm GSD.

The resulting post-processed remote sensing datasets were the photogrammetric point cloud with a 10 cm point interval computed from the RGB image data, the RGB image-based orthophoto mosaics with 5 cm GSD, and FPI orthomosaic with 33 spectral bands with 10 cm GSD.

*2.2. Dataset Preparation for Classification*

In total, 3896 trees were selected in Vesijako blocks v01, v02, v05, v06, v07, v08, v09, and v10. The dataset contained the three most common tree species of Finnish forests: Scots pine, Norway spruce, and silver birch. Figure 2 shows the location and distribution of tree species in the third region. The locations of trees in this area were originally collected by GPS and manually improved using the high resolution RGB orthoimages as described in [7].

For all the proposed models, 3093 tree samples were selected randomly for training and 803 different trees for testing the network. In the training dataset the number of pines, spruces and birches were 2001, 626, and 466 samples, respectively. In this study no augmentation method was performed; all data samples were manually labeled.

For each treetop location, the RGB image was cut around the treetop location in a square shape of $25 \times 25$ pixels in size. This was repeated with the CHM and all spectral channels to form the data cube of each individual tree. The data cubes were saved in a MATLAB 4-dimensional array for the study area. The MATLAB arrays are in the size of $n \times x \times y \times d$, where $n$ is the number of trees, $x$ and $y$ are the length of image patches (here 25 pixels), and $d$ is the number of used data sources (3 RGB, 1 CHM

and 33 HS channels). Here, *d* can vary from 1 to 37. Figure 3 shows an example of treetop images in 37 different channels. In order to find out the most efficient combination of features that results in the best classification accuracy, each feature layer (RGB, HS, CHM) was independently classified by a 3D-CNN model; moreover, all combinations have been individually tried out. The blue channel of the RGB layer was independently considered as a feature layer to emphasize its importance.

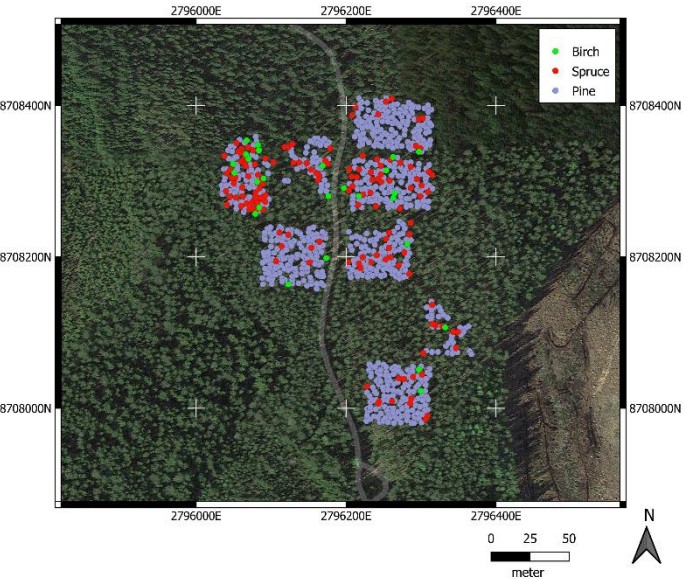

**Figure 2.** Location and distribution of trees in the third region. Each class is individually colored.

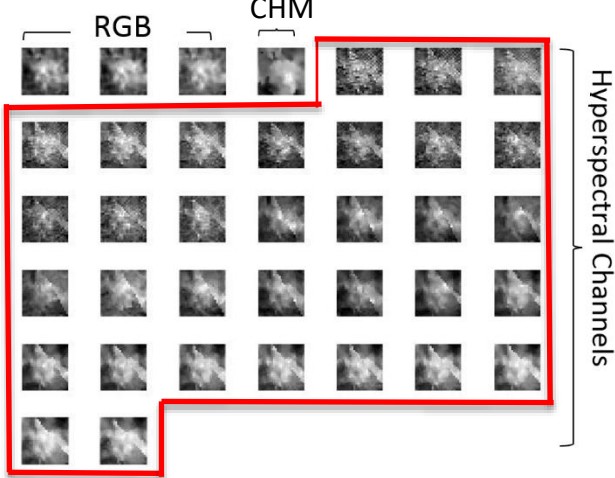

**Figure 3.** Treetop shape in 37 normalized RGB, canopy height model (CHM), and spectral channels.

## 2.3. Classification Models

### 2.3.1. Neural Network (Multi-Layered Perceptron)

A multi-layered perceptron is a feed-forward neural network that is a good candidate for many classification problems. Based on universal approximation theory proved by Cybenko [30] and Hornik [31], MLP can approximate any smooth function in a subset of $\mathbb{R}^n$. An MLP is a fully connected multi-layered feed-forward network that contains an input layer, one or more hidden layers, and one

output layer. The input layer receives the signal and has a number of nodes equal to the dimensionality of the input dataset. In the hidden layer a single node gets $x$ as an input to the combined function

$$f(x) = g\big(w^T x + b\big), \tag{1}$$

where $g(\cdot)$ is the nonlinear activation function such as logistic sigmoid $g(a) = (1 + e^{-a})^{-1}$ or hyperbolic tangent $g(a) = tanh(a)$, $w$ is the weight of the neurons and $b$ is the additional bias. Taking into account all the nodes in a single layer, function can be expressed in matrix form

$$g\big(W^T x + b\big), \tag{2}$$

where $W$ is the weight matrix $W = [w_1, w_2, \ldots, w_i]$ and $i$ is the number of nodes in the layer.

The hidden layers are the main computational core of the MLP and the output layer makes the decision or prediction of input. In our simple two-layered MLP, the output layer is activated with linear transfer function $h(a) = a$. Now, a two-layered network has the following structure:

$$F(x) = h\big(W_2^T g\big(W_1^T x + b_1\big) + b_2\big) = \hat{y}, \tag{3}$$

where it $\hat{y}$ is the classification result for input $x$. For each layer $i$ there are weights $W_i$ and biases $b_i$. Note that $W_i$ is now $[xyd \times l]$ matrix, where $l$ is the number of layers.

The training dataset that contained all 37 layers (RGB+HS+CHM) was employed to train the MLP. The training process involved adjusting the weights $W_i$ and biases $b_i$ of the model to optimize the classification cost

$$\min_{(w,b)} \|F(x) - \hat{y}\|. \tag{4}$$

The trained network was then employed to classify the test dataset [20]. The structure of the MLP used in this study is illustrated in Figure 4. The input layer of the MLP consisted of $25 \times 25 \times 37 = 23125$ nodes with one hidden layer with 10 nodes and an output layer with three nodes. A big challenge in dealing with an MLP is the overfitting problem of the network, which can significantly affect the classification of the test dataset. To avoid overfitting, a small part of the training dataset (~10%) was labeled as the cross-validation dataset. The scaled conjugate gradient optimization method was used to train the MLP. The optimization algorithm stops if any of the following conditions occur: 1: Maximum number of epochs reaches, 2: maximum allowed computation time passes, 3: the cost function decreases below a threshold, 4: the classification performance on the validation dataset sequentially decreases for a defined number of times. The validation dataset triggers the last stopping mechanism.

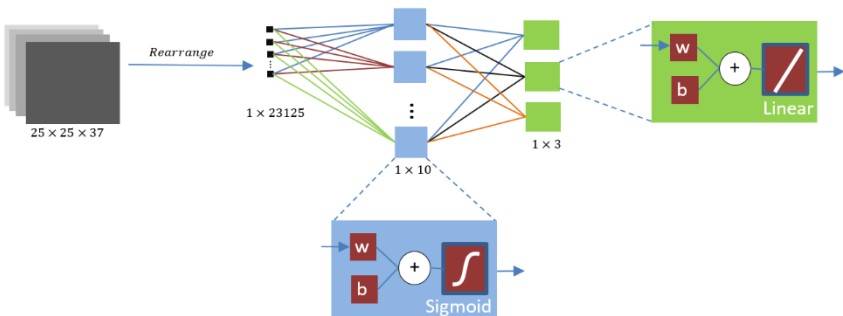

**Figure 4.** Feed forward neural network structure. Hidden and output nodes are magnified as blue and green boxes respectively.

### 2.3.2. 3D Convolutional Neural Network

Convolutional neural networks are related to MLPs. CNNs contain two parts: feature extraction and classification. The feature extraction part consists of convolution filters and pooling operations.

CNNs and MLPs are very similar in nature, since they are both frequently employed to estimate highly non-linear functions of any type. MLPs could be employed as a sub-part of a CNN structure. The main difference between CNNs and MLPs relates to the existence of convolution and pooling in CNNs that plays an existential role to overcome the curse of dimensionality. The other main difference relates to the dense structure of an MLP. This setting significantly increases the number of parameters in an MLP. In a CNN, in contrast, the number of parameters is manageable by considering convolution filters that gradually reduces the data dimension by creating features that best discriminate designed classes. During the training process, weights of the convolution filters are tuned. Three-dimensional discrete convolution is a function of the form

$$(I * K)(x, y, d) = \sum_{l=1}^{h} \sum_{j=1}^{t} \sum_{i=1}^{k} I(x + i, y + j, d + l)K(i, j, l), \tag{5}$$

where $I(x, y, d)$ is original data cube and

$$K(i, j, l) = \begin{bmatrix} w_{11} & \cdots & w_{k1} \\ \vdots & \ddots & \vdots \\ w_{1t} & \cdots & w_{kt} \end{bmatrix} \tag{6}$$

is 3D convolutional kernel with weight vectors $w \in \mathbb{R}^h$. Several different pooling operators exist. Here we used maximum pooling, which selects $s(a) = \max I(x_{ijl})$, while $ijl \in \mathbb{N}^{k' \times t' \times h'}$ and $k'$, $t'$, and $h'$ are the desired pooling kernel dimensions. Both operators work with the sliding window principle going through the whole image. The dimensionality of the input data is gradually reduced after passing through this part [32]. The classification part consists of fully connected layers or convolutional layers to classify the extracted features.

Also, CNNs use the activation function. A commonly used activation function is the rectified linear unit function (ReLU)

$$r(a) = \begin{cases} a, & \text{when } a > 0 \\ 0, & \text{while } a \leq 0 \end{cases}. \tag{7}$$

ReLU is commonly used with gradient-based optimization, because it seems to avoid problems with vanishing gradient. Installing a ReLU layer before the last convolution layer is important, because the ReLU activation function increases the nonlinear properties of the model and the overall network without affecting the receptive fields of the conv layer. These layers form a similar function as in the case of MLP

$$H(I) = s(r(I * K)(I))), \tag{8}$$

which forms a basis for the optimization problem. Deep CNN follows the same structure

$$D(I) = (H_i \circ H_{i'} \circ \ldots \circ H_{i''})(I) = \hat{y}, \tag{9}$$

where $H$'s can have a different number of convolutional kernels. It is possible to combine CNN D(I) to extract features for the MLP $F(x)$ so that

$$F(D(I)) = \hat{y}. \tag{10}$$

Combining is usually done by flattening the output of the convolutional part to a single feature vector, which is then passed to the MLP.

To speed up the actual training process we use batch normalization layers, which normalize the output of the previous layer. In batch normalizations output

$$\frac{a - \mu_B}{\sqrt{\sigma_B^2 + \varepsilon}},\tag{11}$$

where $a$ is an input, $\mu_B$ is the mean, and $\sigma_B^2$ is the variance over the batch of data.

In contrast to an MLP, in a CNN there is no need to connect every node of a given layer to nodes in the next layer. Thus, the number of parameters will be significantly reduced.

The proposed 3D-CNNs have three convolution layers, three batch-normalization layers, two max-pool layers, one rectified linear unit (ReLU), and one Softmax layer. The configuration of the models can be seen in Figure 5. Table 2 lists kernel and output sizes of each layer. It is obvious from this table that the input data shrink as we proceed through the model. The structure of the models is fairly simple, because they are designed in a way to gradually decrease the dimensionality of the input data and ultimately perform the classification task.

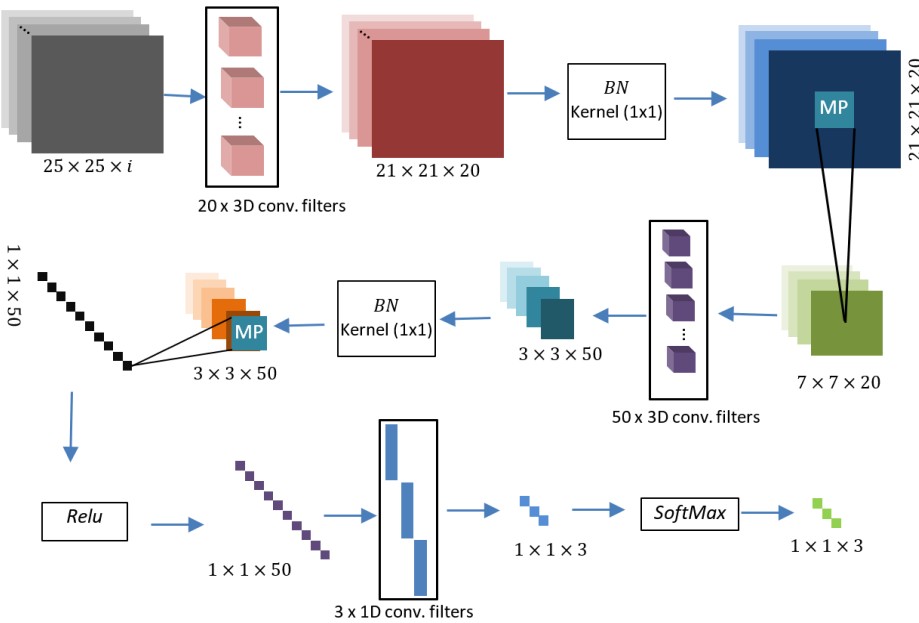

**Figure 5.** Structure of a 3D-convolutional neural networks (CNN) model for input of (i) layers of size $25 \times 25$.

**Table 2.** Configuration of a 3D-CNN model for (i) input layers of size $25 \times 25$.

| Layer | Kernel Size | Kernel Number | Stride | Output Size |
|---|---|---|---|---|
| Input | | - | - | $25 \times 25 \times (i)$ |
| Conv1 | $5 \times 5 \times i$ | 20 | 1 | $21 \times 21 \times 20$ |
| Max Pool | $3 \times 3$ | 1 | 3 | $7 \times 7 \times 20$ |
| Conv2 | $5 \times 5 \times 20$ | 50 | 1 | $3 \times 3 \times 50$ |
| Max Pool | $3 \times 3$ | 1 | 3 | $1 \times 1 \times 50$ |
| ReLU | | - | - | |
| Conv3 | $1 \times 1 \times 50$ | 3 | 1 | $1 \times 1 \times 3$ |
| Soft Max loss | | - | - | 3 |
| Total parameters: | | 43650 (for 37 layers) | | |

Installing a ReLU layer before the last convolution layer is important because ReLU activation function increases the nonlinear properties of the model and the overall network without affecting

the receptive fields of the conv layer. Finally, after applying three convolutional layers, two max pooling and ReLU layers, the Soft max function, an activation function and a core element in deep learning classification outputs a vector that represents the probability distributions of a list of potential outcomes (three different classes).

*2.4. Performance Assessment*

The purpose of this study is to investigate the applicability of 3D-CNNs to classify tree species from HSIs, therefore, evaluating the suggested network models is an essential part of the study. Here we used overall classification accuracy to measure the performance of a given model as

$$OA = \frac{TP + TN}{n_s},$$

(12)

where OA is the overall accuracy, TP is true positive, TN is true negative, and $n_s$ is the total number of samples. The overall accuracy (Equation (12)) could be a slightly misleading indicator in the condition that a training dataset contains classes with an unequal number of samples. In such cases, three more classification quality indicators are important: confusion matrix, producer's accuracy, and user's accuracy. The producer's accuracy is the number of correctly identified samples of a given class divided by the total number of samples of that class. The user's accuracy on the other hand divides the number of correctly identified samples of a given class by the number of samples that have been labeled by the classifier as a given class. Table A1 lists the symbols of the confusion matrix for a 3-class classification problem. In this table, $TP_i$ is the number of true positives of class (i), and $P^i_j$ is the number of samples that truly belong to class (i) but labeled as class (j). The user and producer accuracies are calculated as

$$PA_i = \frac{TP_i}{TP_i + \Sigma_j P^j_i},$$

(13)

$$UA_i = \frac{TP_i}{TP_i + \Sigma_j P^i_j},$$

(14)

where $PA_i$ is the producer accuracy for class (i) and $UA_i$ is the corresponding user accuracy.

The area under the curve (AUC) of the receiver operating curve (ROC) is widely considered as a suitable performance measure for a classifier (see e.g., Bradley [33]). ROC is calculated as a function whose variable is a threshold in [0 1]. This threshold is employed to assign a specific class number to the probability of an item. Its output is the percentage of true positives. The AUC of ROC relates to the probability that a randomly chosen true positive is correctly selected with a higher chance than a randomly chosen negative [33].

## 3. Results

This section presents the results for classifications with MLP and 3D-CNN, as well as comparing different features. The first subsection demonstrates the MLP classification results. This part is followed by the proposed 3D-CNN classification results.

*3.1. Multi-Layer Perceptron*

The optimization process of the MLP is demonstrated as a convex optimization problem in Figure 6. This figure shows the mean square error (MSE) of the predicted labels as a function of the number of epochs. This figure is a general demonstration of the training process of the MLP. It is depicted in this figure that the validation set triggered the stopping criterion after 82 epochs.

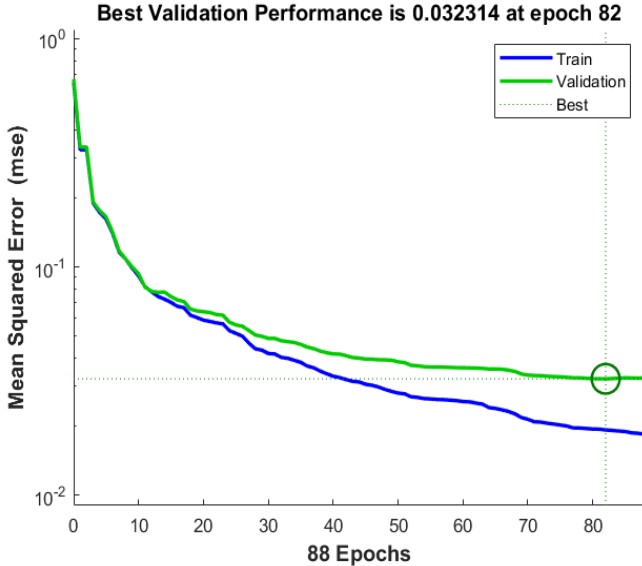

**Figure 6.** The optimization process of the feedforward network.

Figure 7 shows the ROC of the MLP on all data layers (CHM+HS+RGB). AUCs of the MLPs ROC were 0.9961, 0.9590, and 0.9685 for pine, spruce, and birch respectively. This figure demonstrates that the trained MLP is a high-performance classifier.

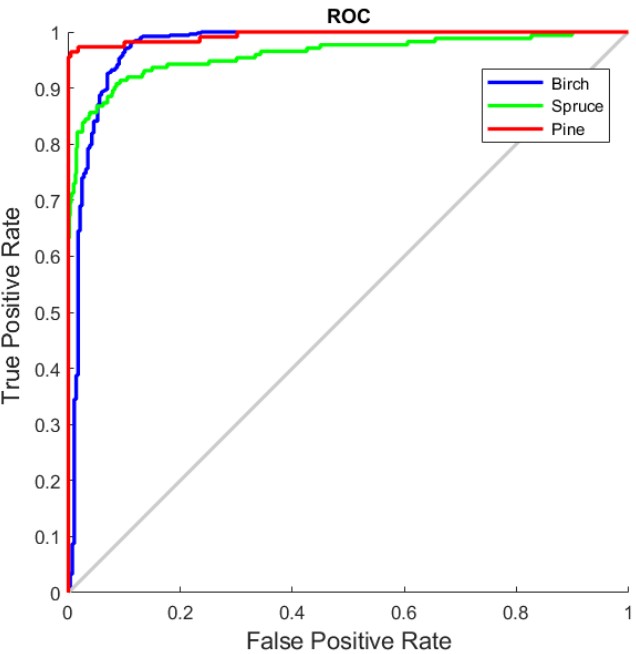

**Figure 7.** Receiver operative characteristic (ROC) curve of the feedforward network for all layers.

Figure 8 shows the confusion matrix of the MLP with all 37 input layers. The overall accuracy of the MLP on the test set of 803 samples was 94.5%. Relatively high accuracies were obtained by employing the MLP. The classification accuracies of the MLP on the training dataset were 98.9%, 90.7%, and 98% for pine, spruce, and birch, respectively. Those accuracies were reduced to 98.4%, 82.2% and 95.6% for pine, spruce, and birch, respectively with the test dataset. The MLP demonstrated an acceptable fitting, since the overall classification accuracies on both training dataset and test dataset were close. Classification with the MLP was relatively quick and efficient in terms of computational time.

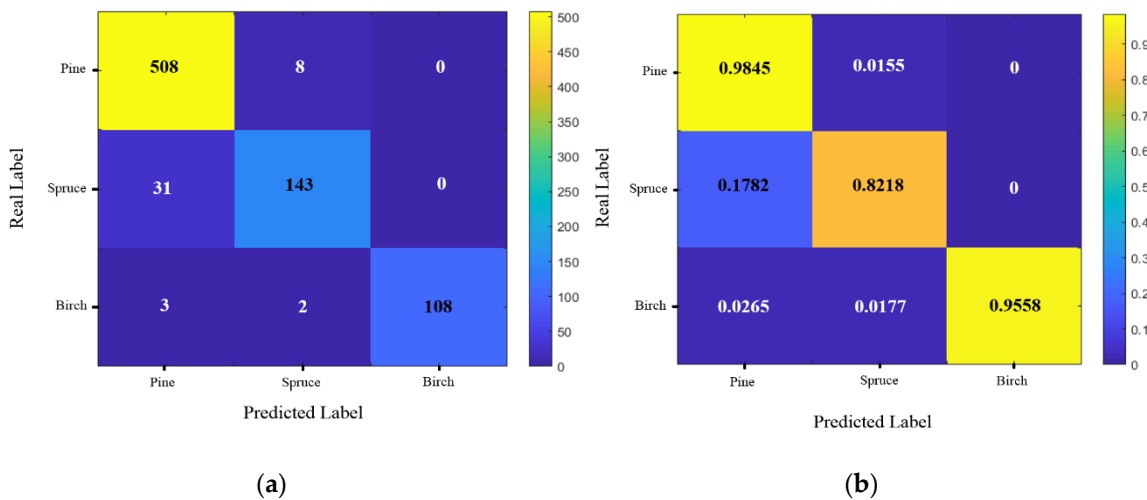

**Figure 8.** Confusion matrices of the multi-layer perceptron (MLP) with all 37 input layers (HS+RGB+CHM); (**a**) case numbers, (**b**) normalized percentages.

## 3.2. Convolutional Neural Network

The proposed 3D-CNN models were trained and evaluated using RGB, HS bands, and CHM separately as well as in different combinations (Table 3). Training was performed with a batch size of 1000 samples during 5000 epochs. The size of the parameter file of the model with all (37) layers was 169 megabytes. The chosen method for training was stochastic gradient descent. Training of the model with all 37 layers took approximately 2.5 h with a computer equipped with a core i7-6820HQ processor with 32 GB random access memory (RAM). Training time for the MLP was considerably faster (16 s). Results were validated with a test dataset of 803 samples, which were not included in the training process. The models were implemented by the MatConvNet library [34], which is a MATLAB toolbox for training CNN models.

**Table 3.** Summary of the overall, producer's, and user's accuracies.

| Feature Set | Producer's Accuracy | | | User's Accuracy | | | Overall Accuracy |
|---|---|---|---|---|---|---|---|
| | Pine | Spruce | Birch | Pine | Spruce | Birch | |
| HS | 0.990 | 0.910 | 0.970 | 0.970 | 0.952 | 1.000 | 0.970 |
| RGB | 0.986 | 0.959 | 0.920 | 0.977 | 0.943 | 0.990 | 0.971 |
| CHM | 0.965 | 0.184 | 0.000 | 0.665 | 0.593 | 0 | 0.660 |
| HS+RGB | 0.996 | 0.948 | 0.974 | 0.981 | 0.976 | 1.000 | 0.983 |
| HS+CHM | 0.99 | 0.897 | 0.965 | 0.964 | 0.951 | 1.000 | 0.966 |
| HS+Blue | 0.986 | 0.920 | 0.974 | 0.971 | 0.947 | 1.000 | 0.970 |
| RGB+CHM | 0.994 | 0.960 | 0.912 | 0.975 | 0.971 | 0.981 | 0.975 |
| HS+RGB+CHM | 0.986 | 0.943 | 0.982 | 0.979 | 0.953 | 1.000 | 0.976 |
| HS+RGB+CHM(MLP) | 0.984 | 0.822 | 0.956 | 0.937 | 0.935 | 1.000 | 0.945 |

Figure 9 demonstrates the ROC of the 3D-CNN model on all data layers with AUC values of 0.9999, 0.9941, and 0.9956 for pine, spruce, and birch, respectively. The proposed 3D-CNN models were trained and tested using the RGB bands, spectral bands, and CHM band separately, as well as different combinations of bands. A summary of the results is listed in Table 3, where the producer accuracy, the user accuracy, and the overall accuracy of each tree type is listed for each model.

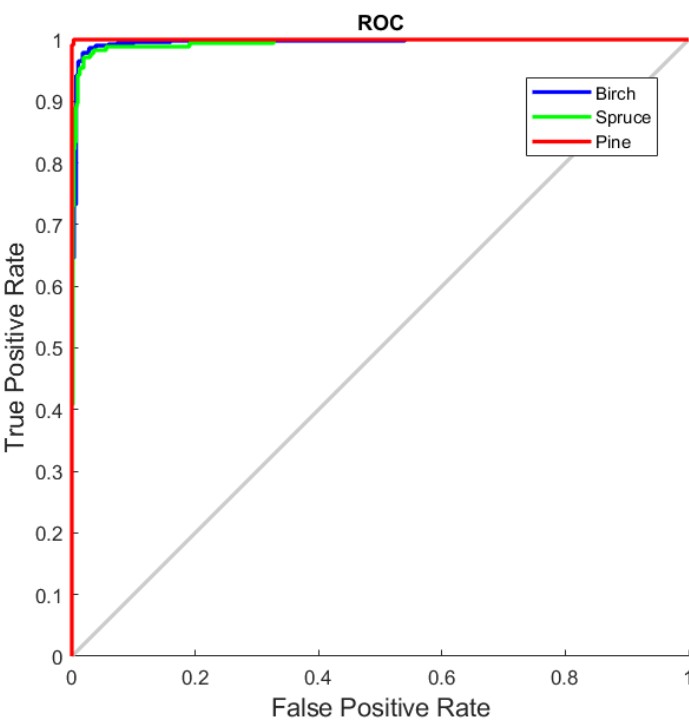

**Figure 9.** Receiver operative characteristic (ROC) curve of the 3D-CNN model for all layers.

Figure 10 shows the confusion matrix of the 3D-CNN model with all 37 input layers. Figures 11 and 12 demonstrate the confusion matrices of the models with HS+CHM layers, and RGB+CHM layers respectively. For (RGB+CHM) model, the accuracies were 99.4% for pine, 96% for spruce, and 91.2% for birch.

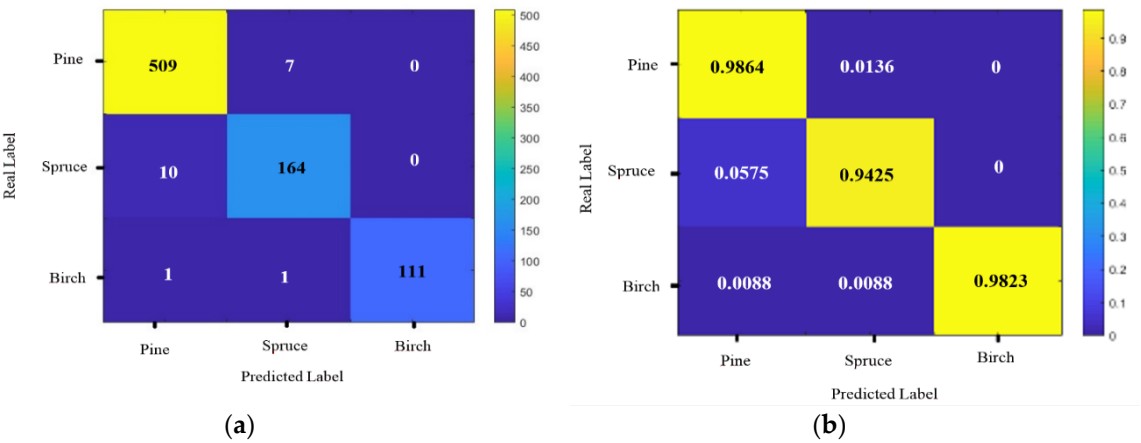

**Figure 10.** Confusion matrices of the 3D-CNN with all 37 input layers (HS+RGB+CHM); (**a**) case numbers, (**b**) normalized percentages.

The confusion matrices of the models with RGB+HS layers are demonstrated in Figure 13. For the (RGB+HS) model, the producer's accuracy was 99.6% for pine, 94.8% for spruce, and 97.4% for birch in the test dataset, and the user's accuracies were 98.1%, 97.6%, and 100% for pine, spruce, and birch, respectively.

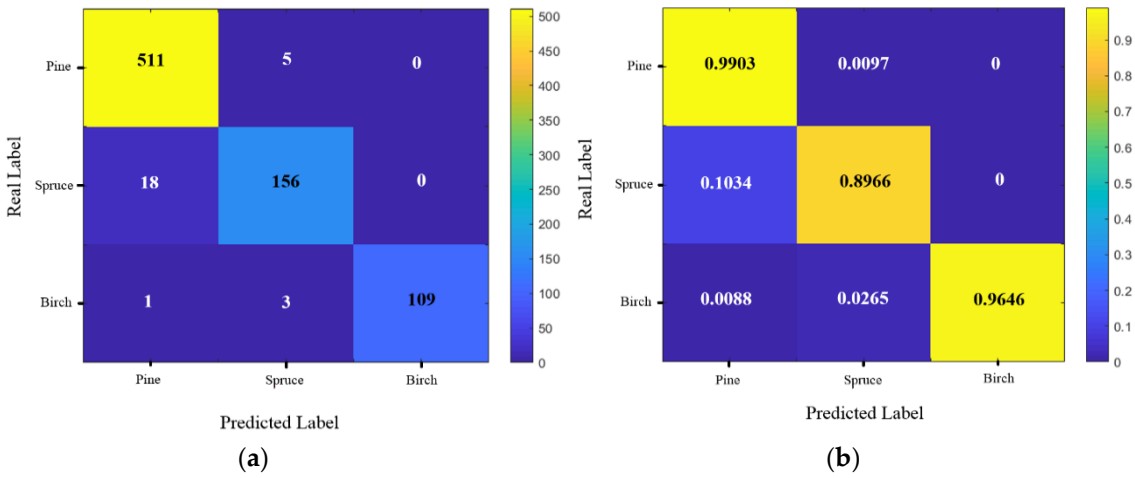

**Figure 11.** Confusion matrices of the 3D-CNN with CHM and spectral channels; (**a**) case numbers, (**b**) normalized percentages.

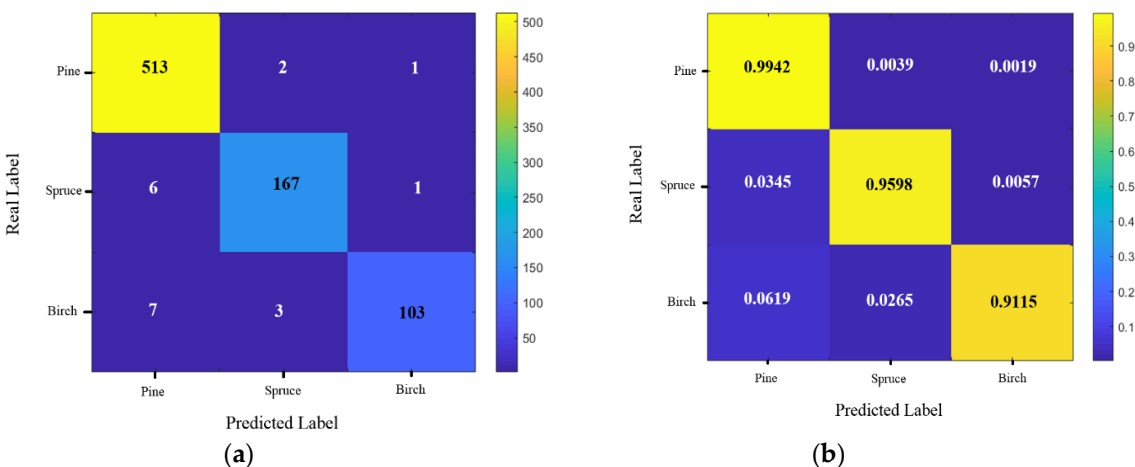

**Figure 12.** Confusion matrices of the 3D-CNN with RGB and CHM channels; (**a**) case numbers, (**b**) normalized percentages.

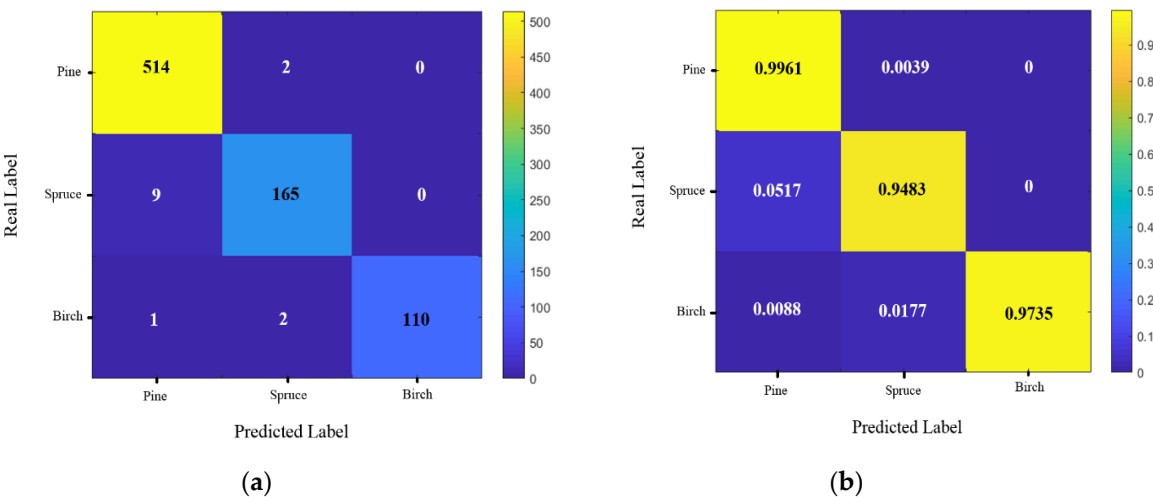

**Figure 13.** Confusion matrices of the 3D-CNN with RGB and spectral channels; (**a**) case numbers, (**b**) normalized percentages.

The producer's accuracy for pine class in Table 3 demonstrates similar results for all the cases. User accuracies for pine are almost similar except for the CHM feature. Almost all the classification results for pine were acceptable, except for CHM layer.

The confusion matrices of spectral-only and RGB-only models are demonstrated in Figures 14 and 15, respectively.

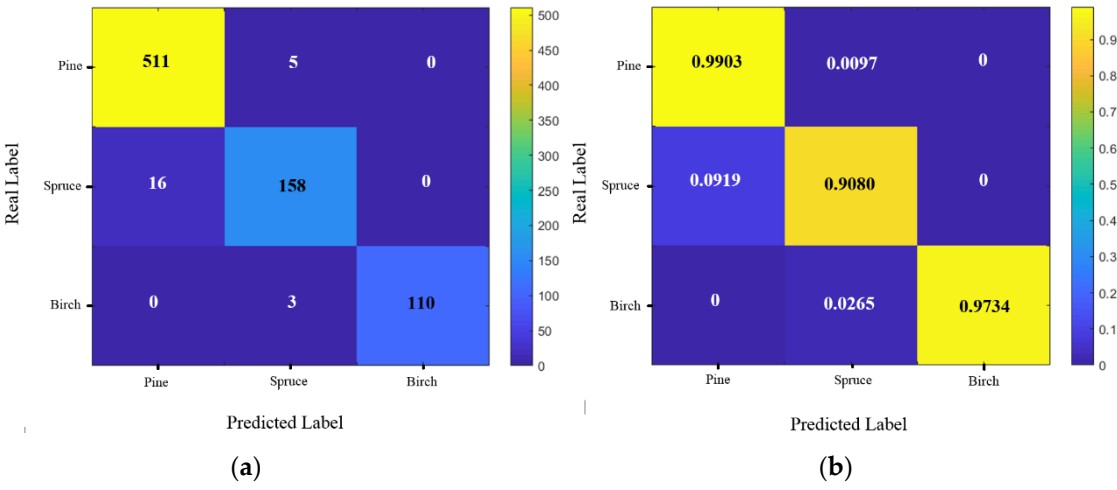

(a)      (b)

**Figure 14.** Confusion matrices of the 3D-CNN with only spectral channels (33 bands); (**a**) case numbers, (**b**) normalized percentages.

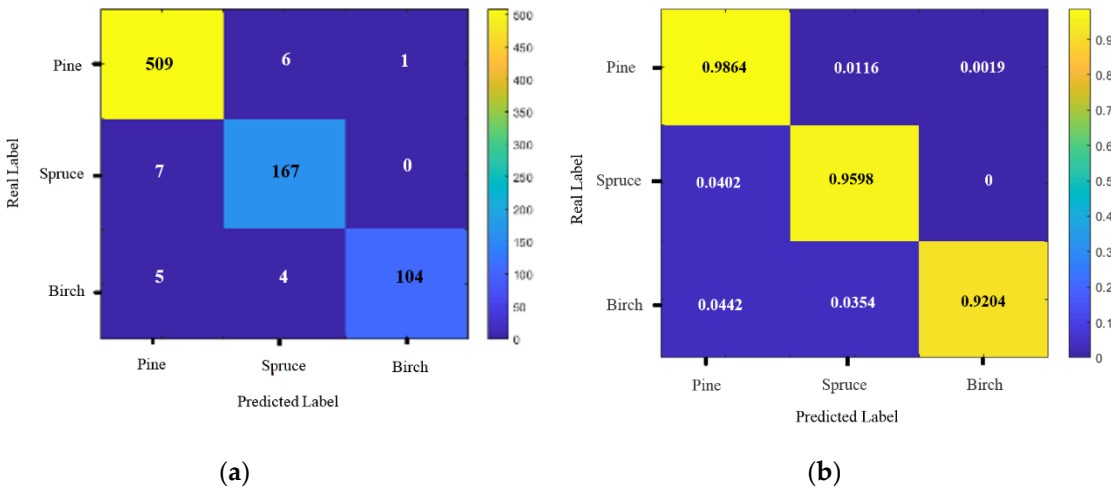

(a)      (b)

**Figure 15.** Confusion matrices of the 3D-CNN with only RGB channels; (**a**) case numbers, (**b**) normalized percentages.

The overall accuracy of HS+Blue was 97% compared to 98.3% of HS+RGB. Birch was classified equally well in both classifications. However, the producer's accuracies for pine and spruce were approximately 2% lower than HS+RGB.

## 4. Discussion

The results show that the 3D-CNN model based on HS and RGB layers provided the best results among all the proposed models. The overall accuracy of the best model (3D-CNN on HS+RGB) was 98.3%, and accuracies of individual classes were 99.6%, 94.8%, and 97.4% for pine, spruce, and birch, respectively.

Both CNN and MLP classifiers successfully performed the classification task. However, the proposed 3D-CNN outperformed the MLP in terms of classification accuracies and number of

parameters. The improvement in classification accuracies was also visible by comparing ROC curves and their respective AUC. The ROC curve of the best 3D-CNN model demonstrated superior results when compared to the MLP model with HS and RGB layers. Between all the tree classes, the ROC of the 3D-CNN demonstrated a greater improvement for spruce and pine classes compared to the MLP results. The decrease in the number of parameters improved the classification time and the amount of required RAM; however, the MLP was quicker after the training.

The model with all layers (HS+RGB+CHM) and the model with RGB and CHM (RGB+CHM) resulted in very close overall classification accuracies to the best model with HS and RGB layers. The RGB-only model provided also very good results compared to the RGB+HS model; the greatest difference was observed in birch, where HS+RGB was approximately 5% better. The overall accuracies, however, were very similar between these two classifiers; the difference was approximately 2%. This result shows that the RGB-only model fits many classification applications. The HS-only model (Figure 14) showed good performances in detecting spruce and birch. However, it had a lower producer accuracy in detecting spruce compared to the models that include the RGB feature.

For pine, the most accurate model was with HS and RGB layers with a producer's accuracy of 99.6%. Classification accuracies of pine were almost similar between all models. For spruce, the better than 94% producer's accuracy was obtained when RGB layers were employed; if RGB layers were not employed, the accuracy was 91% at best. This result highlights that spruce was more visible in RGB layers than in other layers. A classification with HS+Blue layers was performed to test the hypothesis that the blue layer contains a significant signature especially for separating spruce from pine; the blue layer was taken from the RGB dataset as the hyperspectral camera operated in the spectral range of 500–900 nm. The HS+blue model did not meaningfully improve the HS-only model results for spruce (difference ~1%). Also, it showed relatively lower accuracies than HS+RGB, which rejects the stated hypothesis. One possible explanation for the importance of the RGB layers could be the potentially better signal-to-noise-ratio (SNR) of the wide spectral band data of the regular RGB camera compared to the HS data with narrower spectral bands. Visually, the RGB images appeared to have better SNR than the HS data, but this hypothesis could not be measured quantitatively in this study. For birch, the accuracy of better than 96% was obtained when the HS layers were employed; without HS layers the accuracy was 92% at best. This result is consistent with the expectations that birch is distinguishable from the conifers in the NIR spectral range [16,17].

Our best model (3D-CNN on HS+RGB) showed better results (98.3%) than Pölönen et al. [2] having a 96.2% overall accuracy. Also, the number of parameters in our best model (43'650 i = 37) was significantly lower (97%) than their proposed 3D-CNN model (1'807'939). We also obtained better accuracies for each class. In the study by Nevalainen et al. [7] with the same data, the best overall accuracy was achieved with the RF classifier using combined HS and 3D point cloud features (approximately 95%), and when using 3D features only the overall accuracy was 72%. Thus, slightly better overall accuracies were obtained in this study. The results by Nevalainen et al. [7] gave producer's accuracies of 96%, 91.5%, and 98.1% for pine, spruce, and birch for the MLP, respectively, when assessed with the leave-one-out technique. Our results with the test dataset were slightly better (98.6 %, 94.3%, and 98.2% for pine, spruce, and birch, respectively). It should be noted that these results are not directly comparable because of the differences in the test design and the use of different features. First, we used both validation and test datasets to assess the performance, whereas leave-one-out estimation was used in [1]; the latter typically gives slightly positively biased results. Furthermore, different sampling has been used between the studies, but that is not expected to impact the final conclusions, assuming that the datasets are randomly distributed and representative. In addition, Nevalainen et al. [7] used features that were computed for a set of pixels instead of using all the pixels as features, RGB features were not used, and 3D features were based on point clouds rather than CHM. The best model in this study (3D-CNN with HS+RGB features) provided significantly better classification accuracy than the model presented by [7], and according to the authors' knowledge, the individual tree-level classification accuracies were the best ever presented for the boreal tree species [1,13,14,18–22]. However, the results

are always dependent on the complexity and characteristics of the forest. Therefore, the model should be further developed and tested with different training datasets. Furthermore, the individual tree detection should be included in the procedure.

The (RGB+CHM) model was obviously less capable of detecting the birch class in comparison to the (HS+RGB) model. By comparing model (RGB+CHM) to RGB-only model, a small improvement of 0.4% in the overall classification accuracy was observed. As a result, it seems that the CHM layer did not significantly improve the separability of the classes when compared to the RGB-only model.

Including CHM in most cases did not significantly improve the classification accuracy. This showed that CHM was to some extent an irrelevant layer in this classification, or relevant information needed for separating tree species is already covered by other layers such as HS or RGB. Another reason for poor performance with CHM could be due to the relatively low number of points (grid of $25 \times 25$) that have been employed for the classification. In some cases, adding CHM even decreased the classification accuracy. In cases where CHM improved the classification, the improvements were almost always negligible (<1%). Even though the model with CHM did not detect any birches in the classification, the relatively good accuracy of 66% with CHM was due to the fact that the majority of samples were pines (sample imbalance). Studies such as [1] have already demonstrated lower classification accuracy for point cloud-only features in comparison with a combination of CHM and spectral features. In [1] the lowest accuracy was related to birch, which was relatively indistinguishable from pine. In this regard, our results are consistent with [1].

Our results thus suggested that the wide spectral band RGB camera that has a high SNR and the hyperspectral camera that has a high spectral resolution were the ideal sensor combination for the species classification. It is also important to study if the novel multispectral UAV cameras could replace the HS cameras [35]. Furthermore, our results proposed that the CHM did not provide added value for the species classification. This is also an interesting result, because point cloud generation is a laborious part of the processing chain and the processing could be significantly accelerated if the highest density point clouds are not required. However, tree detection and tree physical parameter extraction are often based on point clouds thus further analysis should be carried out to confirm the roles of different remote sensing data layers in different parts of the complete forest inventory process. Further research is thus needed to develop the most efficient procedures.

## 5. Conclusions

In this paper, we proposed a novel 3D-CNN framework for classifying tree species in datasets of color (RGB) images, canopy height models (CHMs), and hyperspectral (HS) images. The proposed models were successfully verified by the classification metrics. The overall accuracy was 98.3% for the best 3D-CNN with HS and RGB bands showing that the proposed model had excellent performance. Moreover, the structure of the 3D-CNN was demonstrated to be efficient in terms of number of parameters in the model. According to the authors' knowledge, our work reported the highest ever presented classification accuracies for three boreal species and suggest that the 3D-CNN, HS, and RGB image-based approaches have the capability to revolutionize the species classification task. Second, we showed the ability of the 3D-CNNs in classifying relatively high dimensional data, where a multi-layered perceptron provided substantially lower accuracies. Our third contribution was the assessment of the performance of different datasets in tree species detection. Ten different classifications were performed: 1: 3D-CNN with a total of 37 bands, including RGB, CHM, and HS images; 2: an MLP with all bands; a 3D-CNN with 3: CHM and HS layers; 4: RGB and HS layers; 5: RGB and CHM layers; 6: only HS layers; 7: only RGB bands; 8: Blue band and HS bands; and 9: only CHM. Results indicated that an improvement of ~5% was achieved for the birch class by using HS+RGB compared to RGB-only classification. On the other hand, by investigating the "only spectral case" (HS layers), we observed a ~5% decrease in classification accuracy of spruce class in comparison with the HS+RGB case. The results from using only the CHM data were poor and the CHM appeared to be an irrelevant layer when combined with RGB and spectral features. Our results suggest that the proposed

model with a combination of RGB and HS images could achieve the best classification result. A further practical implication of this research is the confirmation that an RGB sensor provided excellent results for boreal tree species classification. Our future goals will be to extend the current research on topics such as individual tree detection, developing more efficient 3D-CNN structure and feature selection, extending the model with a wider variety of training datasets, and extending the 3D-CNN model into the complete inventory process.

**Author Contributions:** Conceptualization, S.N. and E.H.; methodology, S.N. and E.K.; software, S.N. and E.K.; validation, S.N., E.K., O.N., and E.H.; investigation, S.N., E.K., O.N., and I.P.; resources, E.H.; writing—original draft preparation, S.N.; writing—review and editing, S.N., E.K., O.N., E.H., and I.P.; visualization, S.N. and E.K.; supervision, E.H.; project administration, E.H; funding acquisition, E.H. All authors have read and agreed to the published version of the manuscript.

**Funding:** This research was financially supported by the Business Finland DroneKnowledge project (Dnro 973 1617/31/2016) and by the Academy of Finland project "Autonomous tree health analyzer based on imaging UAV spectrometry" (Decision number 327861).

**Acknowledgments:** Sakari Tuominen from the Natural Resources Research Institute is acknowledged for the tree species data from the Vesijako test site. Niko Viljanen and Teemu Hakala from Finnish Geospatial Research Institute in National Land Survey of Finland (FGI), are acknowledged for the UAV data capturing and georeferencing. Paula Litkey from the FGI is acknowledged for her help in generating training and testing datasets.

**Conflicts of Interest:** The authors declare no conflict of interest. The funders had no role in the design of the study; in the collection, analyses, or interpretation of data; in the writing of the manuscript, or in the decision to publish the results.

## Appendix A

**Table A1.** Confusion matrix of the classification problem with three classes.

| | | Predicted Labels | | | Total Producer |
|---|---|---|---|---|---|
| | | Pine | Spruce | Birch | |
| Real Labels | Pine | $TP_{pp}$ | $P_p^s$ | $P_p^b$ | $TP_{pp}+P_p^s+P_p^b$ |
| | Spruce | $P_s^p$ | $TP_{ss}$ | $P_s^b$ | $TP_{ss}+P_s^p+P_s^b$ |
| | Birch | $P_b^p$ | $P_b^s$ | $TP_{bb}$ | $TP_{bb}+P_b^p+P_b^s$ |
| | Total User | $TP_{pp}+P_s^p+P_b^p$ | $TP_{ss}+P_p^s+P_b^s$ | $TP_{bb}+P_p^b+P_s^b$ | |

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
