# Peer review of "Tree Species Classification of Drone Hyperspectral and RGB Imagery with Deep Learning Convolutional Neural Networks"

_remotesensing, doi:10.3390/rs12071070_

Round 1
Reviewer 1 Report
I have no more comment to this manuscript.
Author Response
Thanks for your encouraging report. we appreciate the time that you spend on our article.
Reviewer 2 Report
The reviewer is pleased to find that the overall quality of this paper has improved. Most of the comments in the first version have been addressed. As requested, Section 1 has been extended to provide a better motivation for, and a smoother introduction to, the proposed approach. Importantly, the objectives of the study are now stated more clearly (p.3, lines 132-134) and the contributions have been isolated (p.4, lines 141-146) to explain the implications and importance of the study.
The reviewer has several suggestions to further strengthen the statement of contributions, which is a crucial part of this paper:
* "The main contribution of this work includes:" -> "The main contributions of this work are:"
* The bullet points 1-3 would read more concisely if they do not start with verbs in the present participle, e.g., "... evaluating the proposed model..." -> "An evaluation of the proposed model" or "We evaluate the proposed model".
* Bullet point 1, "proposing and investigating an efficient structure for a 3D-CNN" might be better replaced with "introducing an efficient structure for a 3D-CNN based on..." + some distinguishing characteristics.
* "Performing the classification" can be removed.
Section 2, describing the methodology, has also been improved with more in-depth explanations and smoother transitions between sub-sections. The new Figures 1 and 2 are both interesting and useful. The reviewer's main outstanding concern with the paper, however, is that the contribution point 1 above, relating to the proposed 3D-CNN, has not been sufficiently emphasized. For instance, the caption of Figure 5 on p.9 and the associated description of the architecture (p.8, lines 306-307) should state very clearly what is new about this design (compared to previous work). Similarly, in Sections 3 and 4, the authors should connect and discuss their findings to particular aspects of the 3D-CNN structure, in order to validate their claims in the experimental setup. These changes are necessary if the 3D-CNN design is to be listed as a contribution point.
The separation of the results and discussion in Section 4 greatly benefits the paper. In general, the experimental findings are much better supported as well as more thorough. Some wider points for discussion that would be interesting to discuss: What implications do the results have on practical systems, e.g., sensor selection? Can the results by explained by looking at the physical/qualitative appearance of the trees?
The general readability of the paper has been also refined. The authors are to be commended for their careful attention to detail in the proofreading. There are some issues in formatting, e.g., equation above figure and figure above text on p.7, different text fonts/sizes, spilling font in Table 3, accidental new paragraph on p.10, lines 356-358, and layout issues, that need attention. Also, the presentation of the new Figures 6,7, and 9 is rather poor (blurriness and font size) and should be improved for publication quality.
In summary, as a key point before final acceptance, the authors are advised to refine and reinforce the contributions of the study. This can be done by forming a tighter link between the contribution points and the remainder of the paper.
Author Response
The reviewer is pleased to find that the overall quality of this paper has improved. Most of the comments in the first version have been addressed. As requested, Section 1 has been extended to provide a better motivation for, and a smoother introduction to, the proposed approach. Importantly, the objectives of the study are now stated more clearly (p.3, lines 132-134) and the contributions have been isolated (p.4, lines 141-146) to explain the implications and importance of the study.
Answer:
Thanks. We wish the rest of your concerns will be appropriately addressed.
--------------------------------------------------------------------------------
The reviewer has several suggestions to further strengthen the statement of contributions, which is a crucial part of this paper:
1- * "The main contribution of this work includes:" -> "The main contributions of this work are:"
Answer:
The editor’s suggestion was applied.
--------------------------------------------------------------------------------
2- * The bullet points 1-3 would read more concisely if they do not start with verbs in the present participle, e.g., "... evaluating the proposed model..." -> "An evaluation of the proposed model" or "We evaluate the proposed model".
Answer:
All bullet points were edited as the reviewer suggested.
--------------------------------------------------------------------------------
3- * Bullet point 1, "proposing and investigating an efficient structure for a 3D-CNN" might be better replaced with "introducing an efficient structure for a 3D-CNN based on..." + some distinguishing characteristics.
Answer:
This bullet point is edited. The distinguishing characteristic of this work was briefly added here. However, a more precise discussed could be found later in the text.
--------------------------------------------------------------------------------
4- * "Performing the classification" can be removed.
Answer:
This bullet point is removed. A new bullet point is added:
“different feature combinations are compared to find the most relevant feature set”.
--------------------------------------------------------------------------------
5-Section 2, describing the methodology, has also been improved with more in-depth explanations and smoother transitions between sub-sections. The new Figures 1 and 2 are both interesting and useful. The reviewer's main outstanding concern with the paper, however, is that the contribution point 1 above, relating to the proposed 3D-CNN, has not been sufficiently emphasized.
5-A For instance, the caption of Figure 5 on p.9 and the associated description of the architecture (p.8, lines 306-307) should state very clearly what is new about this design (compared to previous work).
Answer:
The proposed structure is discussed in details. The main achievement of this structure is also well stated as classification accuracy and simplicity (lines 145, 455-457),
--------------------------------------------------------------------------------
5-B Similarly, in Sections 3 and 4, the authors should connect and discuss their findings to particular aspects of the 3D-CNN structure, in order to validate their claims in the experimental setup. These changes are necessary if the 3D-CNN design is to be listed as a contribution point.
Answer:
The proposed structure is stated to be more efficient in terms of classification accuracy, and more-compact in terms is number of parameters. As far as I know, this structure led to the best-known classification accuracy for tree species classification. It has 97% less parameters than the previous work.
The separation of the results and discussion in Section 4 greatly benefits the paper. In general, the experimental findings are much better supported as well as more thorough.
--------------------------------------------------------------------------------
6- Some wider points for discussion that would be interesting to discuss: What implications do the results have on practical systems, e.g., sensor selection? Can the results by explained by looking at the physical/qualitative appearance of the trees?
Answer:
We added some wider discussion regarding the sensor selection in the end of the Discussion. Also some discussion is given regarding the spectral characteristics of birches and conifers.
The general readability of the paper has been also refined. The authors are to be commended for their careful attention to detail in the proofreading.
--------------------------------------------------------------------------------
7-There are some issues in formatting, e.g., equation above figure and figure above text on p.7, different text fonts/sizes, spilling font in Table 3, accidental new paragraph on p.10, lines 356-358, and layout issues, that need attention. Also, the presentation of the new Figures 6,7, and 9 is rather poor (blurriness and font size) and should be improved for publication quality.
Answer:
> equation above figure and figure above text on p.7
I can see no equation above figure 4 in page 7.
Table 2 was edited. The “Total parameters” field was missing, so it was refined.
Table 3 was edited. The column labels turned into bold. A column label for “feature set” was added.
Figure 6 seems fine. Figure 7 was edited and magnified a bit for a better visibility. The labels were enlarged for better visibility. Figure 9 was edited and magnified.
The accidental new paragraph was observed on page 10 and fixed.
--------------------------------------------------------------------------------
8- In summary, as a key point before final acceptance, the authors are advised to refine and reinforce the contributions of the study. This can be done by forming a tighter link between the contribution points and the remainder of the paper.
Answer:
We improved the contributions in the Conclusions section and they are now directly linked to the research objectives given in the end of the Introduction section.